# From CGRP to PACAP, VIP, and Beyond: Unraveling the Next Chapters in Migraine Treatment

**DOI:** 10.3390/cells12222649

**Published:** 2023-11-17

**Authors:** Masaru Tanaka, Ágnes Szabó, Tamás Körtési, Délia Szok, János Tajti, László Vécsei

**Affiliations:** 1HUN-REN-SZTE Neuroscience Research Group, Hungarian Research Network, University of Szeged (HUN-REN-SZTE), Danube Neuroscience Research Laboratory, Tisza Lajos krt. 113, H-6725 Szeged, Hungary; tanaka.masaru.1@med.u-szeged.hu; 2Department of Neurology, Albert Szent-Györgyi Medical School, University of Szeged, Semmelweis u. 6, H-6725 Szeged, Hungary; szabo.agnes.4@med.u-szeged.hu (Á.S.); szok.delia@med.u-szeged.hu (D.S.); tajti.janos@med.u-szeged.hu (J.T.); 3Doctoral School of Clinical Medicine, University of Szeged, Korányi fasor 6, H-6720 Szeged, Hungary; 4Faculty of Health Sciences and Social Studies, University of Szeged, Temesvári krt. 31, H-6726 Szeged, Hungary; kortesi.tamas@szte.hu; 5Preventive Health Sciences Research Group, Incubation Competence Centre of the Centre of Excellence for Interdisciplinary Research, Development and Innovation of the University of Szeged, H-6720 Szeged, Hungary

**Keywords:** migraine disorders, headache disorders, nociceptive pain, analgesics, calcitonin gene-related peptide, pituitary adenylate cyclase-activating polypeptide (PACAP), vasoactive intestinal peptide, adrenomedullin, neuropeptides, drug development

## Abstract

Migraine is a neurovascular disorder that can be debilitating for individuals and society. Current research focuses on finding effective analgesics and management strategies for migraines by targeting specific receptors and neuropeptides. Nonetheless, newly approved calcitonin gene-related peptide (CGRP) monoclonal antibodies (mAbs) have a 50% responder rate ranging from 27 to 71.0%, whereas CGRP receptor inhibitors have a 50% responder rate ranging from 56 to 71%. To address the need for novel therapeutic targets, researchers are exploring the potential of another secretin family peptide, pituitary adenylate cyclase-activating polypeptide (PACAP), as a ground-breaking treatment avenue for migraine. Preclinical models have revealed how PACAP affects the trigeminal system, which is implicated in headache disorders. Clinical studies have demonstrated the significance of PACAP in migraine pathophysiology; however, a few clinical trials remain inconclusive: the pituitary adenylate cyclase-activating peptide 1 receptor mAb, AMG 301 showed no benefit for migraine prevention, while the PACAP ligand mAb, Lu AG09222 significantly reduced the number of monthly migraine days over placebo in a phase 2 clinical trial. Meanwhile, another secretin family peptide vasoactive intestinal peptide (VIP) is gaining interest as a potential new target. In light of recent advances in PACAP research, we emphasize the potential of PACAP as a promising target for migraine treatment, highlighting the significance of exploring PACAP as a member of the antimigraine armamentarium, especially for patients who do not respond to or contraindicated to anti-CGRP therapies. By updating our knowledge of PACAP and its unique contribution to migraine pathophysiology, we can pave the way for reinforcing PACAP and other secretin peptides, including VIP, as a novel treatment option for migraines.

## 1. Introduction

Migraines are neurological disorders causing recurrent, severe headaches and other symptoms like sensitivity to light, sound, smell, or touch, and nausea or vomiting [1]. Their cause is unclear but involves genetic, environmental, and lifestyle factors [2,3,4,5]. Triggers vary among individuals and include stress, hormonal changes, certain diets, and sleep disturbances [6,7,8,9]. Identifying and managing triggers can be crucial in preventing the onset of migraine attacks and reducing their frequency, duration, and severity [10]. Migraines are complex neurological disorders that can significantly impact individuals’ quality of life [11,12]. Comprehending the distinct stages, symptoms, triggers, and treatment options is fundamental for healthcare professionals and researchers, as it facilitates enhanced management and support for individuals affected by migraines [10].

Neuropeptides like calcitonin gene-related peptide (CGRP), pituitary adenylate cyclase-activating polypeptide (PACAP), vasoactive intestinal polypeptide (VIP), islet amyloid polypeptide (IAPP)/amylin, substance P, and adrenomedullin (ADM) [13,14,15,16,17,18]. The secretin family of peptides, including CGRP, PACAP, ADM, and amylin, control G protein-coupled receptors (GPCR) activity. They share homology, receptor cross-reactivity, and similar biological actions, suggesting they belong to this family (Figure 1) [19]. These neuropeptides play diverse roles in migraine pathogenesis, contributing to our understanding of the disorder’s mechanisms [20,21].

CGRP and PACAP, two neuropeptides, are released during migraine and cluster headache attacks, acting as potent vasodilators that trigger migraine-like symptoms [22,23,24]. Their expression increases when the trigeminovascular system is activated, contributing to pain signal transmission and the development of mechanical hyperalgesia [25,26,27,28]. Despite their similar functions, PACAP and CGRP likely have distinct roles in causing migraine-like symptoms. In rodent models, their pathways seem to operate independently; therefore, elevated levels of these substances in peripheral blood during migraine attacks may serve as prospective biomarkers [29,30,31,32,33]. Different PACAP variants also contribute uniquely to migraine development [34,35,36,37]. The neuropeptide VIP, found in the trigeminal nerve, plays a key role in the progression of migraines development. It dilates blood vessels during attacks, influences neurotransmitter release, regulates inflammation and immune responses, and may affect migraine intensity and frequency by modulating pain signal sensitivity [38,39,40,41,42].

New drugs targeting the CGRP signaling pathway have been developed for migraine treatment and prevention. These include monoclonal antibodies (mAbs) directed at either CGRP ligand or receptor and CGRP receptor inhibitors [43,44,45,46,47]. While promising, these treatments have some downsides [48,49]. They have been shown to reduce the frequency of migraine attacks: according to reports from double-blind placebo-controlled clinical trials and open-label trials, the ≥50% responder rate for CGRP mAbs ranges from 27 to 62%, or 44.5 to 71.0%, respectively [50,51]. For CGRP receptor inhibitors, the ≥50% responder rate ranges from 56 to 61%, or 44.5 to 71.0%, in double-blind placebo-controlled clinical trials and open-label trials, respectively [52,53]. The relatively higher effective rates of open-label trials compared to double-blind placebo-controlled clinical trials could be attributed to the possibility that placebo plays a role in real life. Nevertheless, the responder rate varies depending on the type of and the duration of treatment, the response criteria, and the patient characteristics. Additionally, these drugs can be costly with limited insurance coverage [54]. While generally well-tolerated, CGRP-targeting mAbs can cause gastrointestinal disorders like constipation, and gepants can cause fatigue, nausea, dizziness, tiredness, and dry mouth [55].

Humans typically experience migraines, but preclinical research using animal models reveals the interaction of genetic and environmental factors contributing to neurological disorders like migraines [56,57,58,59,60,61,62,63,64,65,66,67]. These models simulate disease conditions, aiding in identifying pathogenic processes, evaluating symptoms and comorbidities, and discovering interventions [68,69,70,71,72,73,74]. The integration of preclinical and clinical research contributes to innovative therapeutics and personalized medicine [75,76,77,78]. This review discusses the pathogenesis of migraines and the need for new treatment targets. It highlights the potential of secretin family peptides’ ligands and receptors as novel targets. The importance of further research into the roles of PACAP and VIP in migraine pathophysiology is emphasized, along with the development of targeted therapies. The review also considers the pituitary adenylate cyclase-activating peptide 1 receptor and other emerging therapeutic targets, such as PACAP1–38. It explores the similarities between PACAP and VIP, which are involved in sleep regulation and circadian rhythm, suggesting their key roles in migraines.

## 2. Pituitary Adenylate Cyclase-Activating Peptide and Vasoactive Intestinal Peptide

PACAP is a multi-functional peptide that has therapeutic potential in a variety of pathophysiological conditions and represents a promising avenue for intervention. PACAP is a neuropeptide that plays a crucial role in both neural and endocrine functions [78]. This peptide is widely distributed throughout the body and is involved in diverse physiological processes, including circadian rhythm and immune system regulations, modulation of pain perception, and stress response [79]. PACAP also has neuroprotective effects and has been shown to support nerve cell survival and regeneration in various neurological disorders [80]. GPCRs control the signaling pathways and cause the activation of adenylate cyclase (AC), the release of cyclic AMP, and the activation of protein kinase A (PKA) and calcium channels [81,82]. PACAP is a multi-functional peptide that has therapeutic potential in a variety of pathophysiological conditions and represents a promising avenue for therapeutic intervention [83].

### 2.1. Background

PACAP was found in ovine hypothalamic extracts in 1989. It is a 38-amino acid peptide hormone that stimulates AC activity in the pituitary gland [84]. Subsequently, it was found to be widely distributed in the central and peripheral nervous systems, as well as in non-neural tissues, including the adrenal gland, pancreas, gut, and reproductive system [85]. PACAP exists in three biologically active forms: PACAP1–38, 6–38-amino acid form of PACAP (PACAP6–38), and PACAP1–27 [86]. PACAP-related peptide (PRP) is also a member of the PACAP family [87]. Radioimmunoassay demonstrated that PACAP1–38 levels were approximately 60 times greater than PACAP1–27 levels and 10 times greater than PRP levels [88].

Since its discovery, PACAP has been extensively studied for its potent neuroprotective effects against a diverse range of neurological disorders, including stroke, traumatic brain injury, Parkinson’s disease, and Alzheimer’s disease [89,90]. Recent findings suggest that PACAP may also play a key role in the regulation of immune cell function and cytokine production, highlighting its potential as a therapeutic target for immune-mediated diseases such as rheumatoid arthritis, multiple sclerosis, and asthma [91]. Furthermore, PACAP has been implicated in the regulation of energy metabolism, making it a promising therapeutic agent for the treatment of metabolic disorders such as obesity and diabetes [92]. Overall, the growing body of evidence on the multifunctional properties of PACAP highlights its potential as a novel therapeutic target for a wide range of diseases.

VIP is a 28-amino acid polypeptide that was first characterized in 1970. It is secreted by cells throughout the intestinal tract and is widespread in many internal organs and systems [93]. VIP plays important roles in many biological functions, such as stimulation of contractility in the heart, vasodilation, promoting neuroendocrine–immune communication, lowering arterial blood pressure, and anti-inflammatory and immune-modulatory activity [94]. VIP stimulates the secretion of electrolytes and water by the intestinal mucosa and acts as a neurotransmitter, inducing a relaxation effect in some tissues [95]. VIP is also involved in the pathophysiology of various diseases, including osteoarthritis, cancer, and autoimmune disorders [94]. Furthermore, VIP is implicated in the physiological and pathophysiological roles of migraine [96]. In this context, VIP has been studied for its potential therapeutic applications.

### 2.2. Receptor and Signaling Mechanisms of PACAP and VIP

PACAP plays an important role in a wide range of biological processes such as feeding behavior, stress response, neuroprotection, and regulation of neurotransmitter release. It activates three different GPCRs named PAC1, vasoactive intestinal peptide receptor (VPAC) 1, and VPAC2; these receptors are widely expressed in the central and peripheral nervous systems, endocrine systems, and immune systems [97]. The binding of PACAP to these receptors leads to the activation of multiple signaling mechanisms (Table 1) [98]. 

Activation of the PAC1 receptor by PACAP leads to the activation of the adenylyl cyclase enzyme, which in turn leads to the production of cyclic adenosine monophosphate (cAMP) and the activation of PKA [99]. It also triggers the activation of phospholipase C, which leads to the breakdown of phosphatidyl inositol 4,5-bisphosphate (PIP2) into inositol triphosphate and diacylglycerol (DAG), which activates protein kinase C (PKC) [100]. On the other hand, VPAC1 and VPAC2 receptor activation leads to AC enzyme activation, which leads to the generation of cAMP and the activation of PKA [101]. Also, PACAP signaling turns on calcium signaling, which causes intracellular calcium to be released and calcium/calmodulin-dependent kinase II to be activated [102]. PACAP signaling also activates the mitogen-activated protein kinase (MAPK), extracellular signal-regulated kinase (ERK), and jun N-terminal kinase signaling pathways [103]. These signaling mechanisms contribute to the diverse biological effects of PACAP on cellular functions. The regulation of PACAP gene expression is presented in Figure 2.

PACAP and VIP are neuropeptides that interact specifically with three receptors (VPAC1, VPAC2, and PAC1) from the class II B GPCR family [104]. The similarities between PACAP and VIP in receptor and signaling mechanisms include the following: PACAP and VIP share nearly 70% amino acid sequence identity; PACAP binds with high affinity to all three receptors, while VIP binds with high affinity to VPAC1 and VPAC2 receptors and has a thousand fold lower affinity for the PAC1 receptor compared to PACAP; both PACAP and VIP receptors are preferentially coupled to Gαs, leading to activation of AC, subsequent cAMP production, and activation of PKA; and PKA may in turn activate ERKs, PACAP and VIP receptor-mediated signaling pathways [105,106,107,108]. Due to the wide distribution of VIP and PACAP receptors in the body, potential therapeutic applications of drugs targeting these receptors, as well as expected unwanted side effects, are numerous [109]. Designing selective therapeutics targeting these receptors remains challenging due to their structural similarities.

### 2.3. Role of PACAP and VIP in Migraine

PACAP has been strongly associated with the pathophysiology of migraine. PACAP is found in high levels in the trigeminal nerve, which is known to play a critical role in this condition. PACAP is known to increase the sensitivity of the trigeminal nerve, cause dilation of blood vessels in the brain, and trigger inflammation. All these biological effects have been implicated in the development of migraine attacks [110]. Several studies have been conducted to investigate the role of PACAP in migraine. One study showed that PACAP levels in the blood are significantly higher in migraine patients during an attack compared to headache-free controls [111]. This study suggests that PACAP could be used as a potential biomarker for migraine. Another study demonstrated that the venous infusion of PACAP into migraine patients resulted in the development of migraine-like attacks [112]. This finding strongly supports the hypothesis that PACAP plays a crucial role in the pathophysiology of migraine and suggests that blocking PACAP could be a potential therapeutic target for the treatment of migraines. The role of PACAP in migraine pathology is well established, and there is strong evidence that this neuropeptide plays a crucial role in the development of migraine attacks. Further research is needed to better understand the mechanism of action of PACAP and to develop new pharmacological agents that target PACAP for the treatment of migraines.

Both CGRP and PACAP are multifunctional peptides with many roles in the nervous, cardiovascular, respiratory, gastrointestinal, and reproductive systems. They play a role in vasodilation, neurogenic inflammation, and nociception. While CGRP plays an integral role in migraine, PACAP is likely to play a similar but distinct role as CGRP based on similarities and differences observed in both clinical and preclinical studies [113]. In rodent models, the PACAP pathway appears to be independent of the CGRP pathway, suggesting that CGRP and PACAP act in parallel ways that cause a migraine-like symptom [114]. In migraine without aura, the first double-blinded placebo-controlled study reported that 33% of the patients developed delayed migraine attacks after CGRP administration [115]. The studies have identified the involvement of two endogenous neuropeptides, CGRP and PACAP, in the pathogenesis of migraines [116].

VIP has also been implicated in the pathophysiology of migraine [117]. The similarities between PACAP and VIP in their roles in pathogenesis include the following: PACAP and VIP are released in conjunction with migraine and cluster headache attacks [118]; PACAP and VIP are potent vasodilators and can cause migraine-like attacks when infused into people [119]; a 2-h infusion of VIP caused migraine attacks, indicating that VIP plays a significant role in pathophysiology and intravenous administration of PACAP-38 caused headaches in all healthy subjects and migraine-like attacks in 58% of patients with a history of migraine without aura [15,35]; PACAP and VIP receptors are preferentially coupled to Gαs, leading to activation of AC, subsequent cAMP production, and activation of PKA [120]; PKA may in turn activate ERKs [121]; PACAP and VIP receptor-mediated signaling pathways are shown to share activities, including vasodilation, neurogenic inflammation, and nociception in rodents [122]; PACAP and VIP receptors provide a rich set of targets to complement and augment the current CGRP-based migraine therapeutics; VPAC1 receptors play a dominant role in PACAP-induced vasorelaxation in female mice [123]. Also, PG 99-465, a selective VPAC2 receptor antagonist that has been used in a number of physiological studies, has been shown to have significant activity at VPAC1 and PAC1 receptors [124].

### 2.4. Preclinical Studies

In addition to in vitro systems, a variety of organisms are used in experimental medicine [125,126,127]. Understanding the effects of endogenous neuropeptides, neurohormones, and metabolites has advanced significantly thanks to the information gathered using laboratory animals [128,129,130,131,132,133]. Animal models are a crucial tool for bridging the knowledge gap between data- and hypothesis-driven benchwork and its application to clinical bedside management. PACAP has been extensively studied as a neuromodulator in the trigeminal nociceptive pathway [134]. Preclinical studies have shown that PACAP is involved in the transmission of pain signals from the periphery to the central nervous system and is therefore a potential target for the treatment of migraine and other headache disorders [135,136].

In animal models, PACAP has been shown to play a role in trigeminal sensitization, which is the process by which nociceptive signals become amplified and persistent, leading to chronic pain [137]. Studies have also found that PACAP is involved in the activation of inflammatory pathways in the trigeminal nerve, further contributing to pain and inflammation [138]. In addition, PACAP has been implicated in the regulation of blood flow to the brain, which may also play a role in headache pathophysiology [139] and other neurological [26] or neuropsychological conditions [88]. In an experimental model of migraine, intraperitoneal administration of nitroglycerol caused marked photophobia and meningeal vasodilatation, and increased the number of c-fos-positive activated neurons in the TNC in wild-type mice but not in PACAP1–38-deficient mice [140]. In line with this, an increased concentration of PACAP1–38 was detected in the TNC after the activation of the TS in different animal models [141,142].

PAC1 receptor antagonists include PACAP6–38, N-stearyl-[Nle17] neurotensin-(6-11)/VIP-(7-28), deletion mutants of maxadilan, M65, and Max.d.4, and synthesized small-molecule acyl hydrazides, including PG 97-269 [143]. PACAP6–38 has been used as a PAC1 receptor antagonist in many studies, but it has an affinity for VPAC2 receptors [144]. N-stearyl-[Nle17] neurotensin-(6-11)/VIP-(7-28) (SNV) is a chimeric peptide analog that antagonizes the VIP2/PACAP receptor subclass. SNV is a better mitogen for the keratinocytic cell line and can increase AC activity in rat brain membranes 100 times more than VIP1-28 [145,146]. No migraine-related studies have been documented. The maxadilan is a vasodilator peptide derived from the salivary glands of sandflies. Its deletion mutants, M65 and Max.d.4, have been reported to be selective PAC1 receptor antagonists but have not been extensively used due to problems of availability [147,148]. PG 97-269 is a selective VPAC1 receptor antagonist with negligible affinity for the PACAP1 receptor. It did not stimulate AC activity but inhibited competitively the effect of VIP on AC activity in cells expressing the VIP1 receptor [146]. VIP and PACAP-induced vasodilation were partially blocked by PG 97-269, indicating that PACAP and VIP may play a role in migraine pathophysiology and that PG 97-269 may have therapeutic potential for migraine [149] (Table 2). Thus, preclinical studies suggest that concentrating on the PACAP signaling pathways in the trigeminal nociceptive system could be an effective strategy for discovering novel treatments for headache disorders. However, more research is needed to fully understand the mechanisms underlying PACAPs’ role in headache pathophysiology and to develop effective and safe PACAP-targeted therapies.

VIP plays a key role in sensory processing and the modulation of pain pathways in the trigeminal system. In preclinical studies, VIP has been shown to change the activity of nociceptive neurons in the trigeminal ganglion and make the TNC more sensitive, which can cause chronic pain or migraines [150]. In response to noxious stimuli, the trigeminal sensory neurons release VIP. This can activate VIP receptors on nearby neurons and cause the release of a number of signaling molecules involved in pain amplification [151]. VIP-mediated sensitization of trigeminal neurons can lead to hyperexcitability and increased responsiveness to noxious stimuli, which may contribute to the development and maintenance of chronic pain or migraine [152]. Targeting VIP signaling pathways may therefore represent a promising approach for the development of novel therapies for chronic pain or migraine.

### 2.5. Clinical Studies

A growing body of clinical research suggests that PACAP plays an important role in migraine pathophysiology. Patients with migraines exhibit higher levels of PACAP compared to control groups [153]. PACAP is a neuropeptide recognized for its involvement in the activation of nociceptive pathways, contributing to the development of migraines. The high levels of PACAP in migraineurs have been associated with increased headache severity and frequency, and this has led to the exploration of PACAP as a therapeutic target for treatment [154]. In migraineurs without aura, the development of PACAP1–38-evoked migraine-like attacks was independent of the severity of family load [35,155]. In the same study, 90 min after the injection, the levels of numerous migraine-related molecular markers were increased in the plasma of patients [156]. Magnetic resonance imaging angiography examinations revealed that PACAP1–38-induced headache was associated with prolonged vasodilatation of the middle meningeal artery (MMA) but not the middle cerebral artery (MCA). Sumatriptan, an antimigraine medication, was able to alleviate the headache, which mirrored the contraction of the MMA but not the MCA, indicating that PACAP1–38-induced headaches may originate from extracerebral arteries [157].

An increasing number of clinical studies have shown that targeting PACAP signaling may be a promising therapeutic strategy for migraine treatment. In terms of safety, PACAP has been generally well tolerated in clinical trials [158]. One study found that PACAP induces headaches via sustained vasodilation and that targeting the PACAP pathway may be a promising approach for treatment [159]. AMG 301, a mAb that targets the PAC1 receptor, was administered to patients with episodic or chronic migraines in a randomized, double-blind, placebo-controlled phase 2 study. There was no significant difference between the AMG 301 group and the placebo group, suggesting that AMG 301 was ineffective for prevention [160,161]. On the other hand, the PACAP ligand mAb, Lu AG09222, was shown to reduce the number of monthly migraine days from baseline to weeks 1–4 of treatment statistically significantly more than placebo [162,163]. Additionally, the mAb targeting the PAC1 receptor, LY3451838, is currently undergoing phase 2 clinical trials for adults with treatment-resistant migraine. This trial is in progress, and the results are not yet available [164] (Table 3). Overall, the efficacy and safety of PACAP as a migraine treatment in clinical studies suggest that it is a promising option for patients with this debilitating condition. Further research is needed to fully understand the potential of PACAP as a treatment for migraines, but the current evidence is encouraging.

VIP infusion has been studied in the context of migraines, with a particular focus on its potential to provoke migraine attacks and its role in pathophysiology. A phase 2 clinical trial investigated the effects of a long-lasting infusion of VIP on headaches, cranial hemodynamics, and autonomic symptoms in episodic migraine patients without aura [165]. The study found that a 2-h infusion of VIP promoted long-lasting cranial vasodilation and delayed headaches in healthy volunteers, resembling the effect of prophylaxis. However, other studies have suggested that VIP infusions may actually provoke migraine attacks. For example, a randomized clinical trial found that a 2-h infusion of VIP caused migraine episodes, suggesting an important role of VIP in migraine pathophysiology [15]. It remains unclear whether the lack of migraine induction can be attributed to the only transient vasodilatory response after a 20-min infusion of VIP. Overall, the search results suggest that VIP infusion may have a role in migraine pathophysiology, but further research is needed to fully understand its effects and potential therapeutic applications.

## 3. Discussion

This review paper aims to provide insights into the roles of PACAP in migraine by comparing its actions with those of VIP. By analyzing existing studies, this paper hopes to shed light on the pathophysiology of migraines and pave the way towards more effective treatments. The ultimate goal of this review is to explore the potential of developing antimigraine drugs that target the PACAP pathways. Identifying and producing new ways to target the PACAP system may provide an alternative therapeutic option for migraineurs. The authors aim to consolidate the current evidence on the PACAP system’s role in migraines and evaluate potential drug targets within the pathway, hoping to pave the way for more extensive research to develop new and effective antimigraine drugs that target the PACAP pathways.

The PACAP system presents a significant challenge when it comes to targeted therapies due to its pleiotropic roles in the body, both physiologically and pathologically [78,79,80,81,82]. PACAP plays crucial roles in various aspects of the body, such as neural development, pain regulation, immune functions, and stress responses. These diverse roles make the PACAP system difficult to target effectively without affecting other physiological functions. Furthermore, PACAP signaling is often dysregulated in pathological conditions such as inflammatory disorders, neurodegenerative diseases, and cancers [91,92]. Conversely, PACAP has been shown to have protective effects in certain diseases, such as ischemic stroke and Alzheimer’s disease [89,90]. Thus, finding a balance between targeting the PACAP system to treat diseases while preserving its physiological functions remains a significant challenge in the field of medicine.

The PACAP system has emerged as a potential target for the treatment of migraines, especially after the discovery of the role of CGRP and its receptors in pathophysiology [110,111,112]. PACAP is a peptide that belongs to the family of CGRP peptides and is highly expressed in the TS. The TS is the neural network that causes migraine pain [137,141,142]. PACAP receptors have been found to be co-localized with CGRP receptors in the TS, suggesting that the two systems could be acting in a synergistic manner to induce migraine pain [113,114,115]. Therefore, targeting the PACAP system could provide an additional therapeutic approach for the treatment of migraine, and several drugs that inhibit PACAP or its receptors are currently under development.

The present review holds notable significance in shedding light on the critical role of PACAP in comparison with other neuropeptides like CGRP and VIP, which have been extensively studied as potential therapeutic targets for various neurological disorders. The differences in symptomatic manifestation observed in preclinical studies of CGRP, PACAP, and VIP are most likely due to their distinct roles in migraine physiology and pathophysiology [105,106,107,108,109,113,114,115]. Thus, elucidating the mechanisms of those neuropeptides may not only lead to a better understanding of the etiology of migraine but may also provide a variety of therapeutic targets, potentially supplying a more diverse palette of antimigraine regimens [150]. By thoroughly analyzing the preclinical studies, the review highlights the promising findings that suggest the potential translation of PACAP’s therapeutic benefits from laboratory settings to clinical practice. The authors’ critical evaluation and systematic compilation of the latest research on PACAP is bound to have a relevant impact on the scientific community and serve as a foundation for further clinical research. Ultimately, the knowledge and insights gained from this review will be instrumental in developing advanced treatments for a range of debilitating neurological conditions.

The difference between those two clinical outcomes of PACAP mAbs could be explained by the fact that mAbs are designed to target specific receptors or ligands with high selectivity. The difference in how mAbs target receptors or ligands can result in different outcomes due to a variety of factors. Initially, mAbs can bind to various receptors or ligands in a variety of ways, which can alter their efficacy and the biological effects that follow [166]. Secondly, mAbs can have a variety of mechanisms of action when interacting with their targets, such as inhibiting cell surface receptors or promoting target cell death [167]. Thirdly, biological and clinical activities can vary greatly depending on the target and antibody design. This includes differences in the efficacy of the treatment, the occurrence of adverse effects, and the overall health of the patient [166]. Fourthly, mAbs exhibit exceptional target selectivity, with the choice of target influencing the antibody’s specificity and safety profile. When mAbs interact with their targets, they can perform a variety of actions, such as inhibiting the action of other molecules, killing cells, or altering the immune system’s function [166,167]. The choice of target and antibody design is crucial in determining the therapeutic effectiveness of mAbs.

The review also highlights limitations and challenges in PACAP research, such as the complexity of its signaling mechanism, variations in its effects on different cell types, and the limited availability of specific antibodies against PACAP and its receptors. The high cost of producing PACAP analogs and the lack of standardized protocols for their synthesis and purification are also limitations. The scarcity of studies on PACAP and VIP is also a major challenge for this field. It is difficult to establish a general agreement on the preclinical results and their relevance for human trials. Meta-analyses could be helpful in this regard, but they require more studies to be published. Therefore, more clinical investigations are necessary to gather evidence and, hopefully, derive conclusions from the clinical research. These challenges and limitations make it difficult to fully understand the mechanisms of PACAP action and to develop effective therapeutic interventions.

The development of PACAP-based therapeutics for migraines will focus on two main approaches: targeting PACAP ligands and receptors. Studies using animal models of migraines have demonstrated that blocking the PACAP receptor reduces symptoms while inhibiting PACAP signaling reduces pain sensitivity. Currently, clinical trials are underway to assess the safety and effectiveness of various PACAP-based drugs for migraines in humans. PACAP-based therapies may offer an alternative to current treatments by targeting the underlying mechanisms of the disorder and reducing the risk of side effects. In addition, the role of additional secretin family peptides, ADM, and amylin in the pathogenesis of migraine remains to be investigated. Further research in this area could lead to the development of better treatments for migraines.

The future direction of migraine research holds great promise for advancing our understanding of this complex neurological disorder. The combination of preclinical and clinical data, along with computational tools, has provided invaluable insights into various aspects of diseases, including neurological and psychiatric disorders [168,169,170,171,172,173,174,175,176,177,178,179,180,181,182,183,184,185,186,187,188,189]. The use of preclinical models and clinical studies has shed light on the underlying mechanisms of migraine. These studies have contributed to the identification of structural and functional changes in the brain that occur in neurological and psychiatric disorders, such as migraine attacks, as well as conditions like depression and other mental health problems [190,191,192,193,194,195,196,197,198,199,200,201,202,203,204,205,206]. Understanding these changes is crucial for identifying biomarkers, developing targeted treatments, and improving diagnosis [207,208,209].

Migraine is not just a pain disorder, but it is also interrelated to emotional and cognitive domains [210]. This condition is commonly linked with a broad range of psychiatric comorbidities, especially among subjects with migraine with aura or chronic migraine [211]. The comorbidity between neurological and psychiatric disorders likely suggests multiple causes, such as unidirectional causal explanations or shared environmental and/or genetic risk factors, communication with other parts of the body, and their interaction on multiple levels [212,213,214,215,216,217,218,219,220,221,222,223,224,225,226]. Emotional distress is commonly recognized as a migraine trigger, and being affected by psychiatric disorders is considered an independent modifiable factor of progression toward chronification of migraine and a tendency to overuse medication [227]. Therefore, revealing the mechanisms of comorbidity between migraine and psychiatric disorders may lead to a clue to prevention and management. Many biological and neural aspects of the comorbidity need to be clarified in order to better understand the true nature of the migraine–psychiatric disorder association.

The integration of computational tools in migraine research has allowed for the testing and evaluation of potential treatments. These tools enable researchers to simulate the effects of different interventions, including brain stimulation, and assess their therapeutic efficacy [228,229,230,231,232]. This approach holds promise for the development of novel and more effective treatments. Advanced imaging techniques have played a crucial role in migraine research. Neuroimaging studies have revealed structural and functional brain changes associated with migraine [233,234,235,236,237,238,239,240]. These imaging techniques provide valuable insights into the pathophysiology of the disorder and can help identify unique clinical cases. The use of human brain organoids in migraine research is an emerging area of study. Brain organoids are three-dimensional models that mimic the structure and function of the human brain. They can be used to investigate altered neuronal pathways, protein expression, and metabolic pathways associated with migraines [241,242,243,244]. This approach offers a unique opportunity to study the disease in a more physiologically relevant system.

## 4. Conclusions

PACAP is a neuropeptide that has been linked to the pathophysiology of primary headaches such as migraine. The release of PACAP is associated with this condition and cluster headache attacks, and it has been shown to be a potent vasodilator that dilates cranial arteries and causes migraines when infused into patients. Like CGRP, PACAP is located near sensory nerve fibers and has nociceptive functions. Both peptides are promising targets for migraine therapeutics, and growing evidence supports the involvement of PACAP-related mechanisms in migraines. While CGRP and PACAP share similar functions, the PACAP pathway appears to be independent of the CGRP pathway, suggesting that they act in parallel ways to cause a migraine-like symptom. Therefore, a better understanding of the role of PACAP and other secretin family peptides, including VIP, in migraine pathogenesis could lead to new treatment options for this debilitating condition.

## Figures and Tables

**Figure 1 cells-12-02649-f001:**
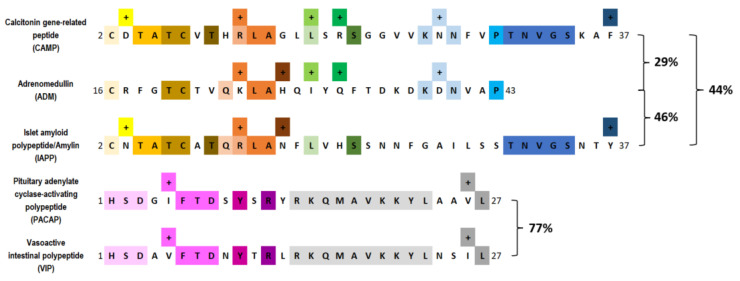
The amino acid sequence alignment analysis of the main secretin family peptides. Those amino acids with matching hues are identical amino acid sequences. The alignment similarity between peptides is displayed as a percentage next to the brackets.

**Figure 2 cells-12-02649-f002:**
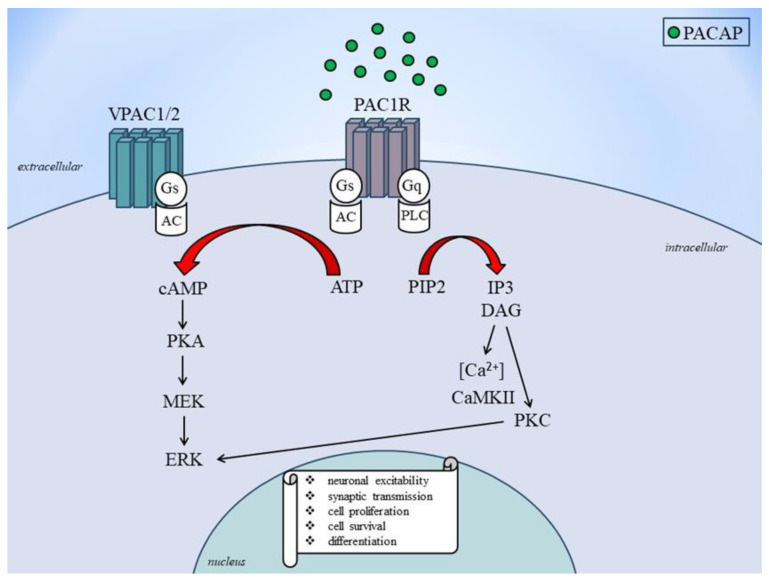
PACAP receptors signaling to ERK activation. AC, adenylate cylase; ATP: adenosine monophosphate; cAMP: cyclic adenosine monophosphate; DAG: diacylglycerol; ERK, extracellular signal-regulated kinase; Gs and Gq: stimulatory G protein; MEK: mitogen-activated protein kinase kinase; PKA: protein kinase A; PKC: protein kinase C; PACAP: pituitary adenylate cyclase-activating polypeptide; PAC1: PACAP 1 receptor; PIP2: phosphatidylinositol bisphosphate; VPAC1: vasoactive intestinal peptide receptor type 1; VPAC2: vasoactive intestinal peptide receptor type 2.

**Table 1 cells-12-02649-t001:** The secretin family peptides, their receptors, and their binding affinity.

Peptides	Receptors
CGRP	CLR
PACAP1–38	>>PAC1, <VPAC1, <VPAC2
PACAP6–38	?
PACAP1–27	>PAC1, <VPAC1, <VPAC2
PRP	?
VIP	>VPAC1, >VPAC2, <PAC1

CGRP: calcitonin gene-related peptide; PACAP: pituitary adenylate cyclase-activating polypeptide: PRP: PACAP-related peptide; VIP: vasoactive intestinal peptide; CLR: calcitonin receptor-like receptor; PAC1: pituitary adenylate cyclase-activating polypeptide type I; VPAC: vasoactive intestinal peptide receptor; ?: unknown; >>: much higher; >: higher; <: lower.

**Table 2 cells-12-02649-t002:** Preclinical findings of PACAP receptor antagonists.

Antagonists	Characteristics	Ref.
PACAP6–38	PAC1 receptor antagonist, affinity for VPAC2 receptors	[144]
N-stearyl-[Nle17] neurotensin-(6–11)/VIP-(7–28)	VIP2/PACAP receptor antagonist, mitogen for the keratinocytic cell line and can increase AC activity	[145,146]
Maxadilan mutants	PAC1 receptor antagonists, increased AC activity	[147,148]
PG 97-269	selective PAC1 receptor antagonists	[146]

**Table 3 cells-12-02649-t003:** Pituitary adenylate cyclase-activating polypeptide (PACAP) monoclonal antibodies under clinical trials.

ClinicalTrials.gov Identifier	Monoclonal Antibody	Target	Status	Ref.
NCT03238781	AMG 301	receptor	No benefit over placebo for migraine prevention	[160,161]
NCT05133323	Lu AG09222	ligand	No results posted; the press release announced a decrease in the number of migraine days per month	[162,163]
NCT04498910	LY3451838	receptor	No results posted	[164]

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
