# Peer review of "From CGRP to PACAP, VIP, and Beyond: Unraveling the Next Chapters in Migraine Treatment"

_cells, 2023, doi:10.3390/cells12222649_

Round 1
Reviewer 1 Report
Comments and Suggestions for Authors
This paper is useful for the reader. Needs bit more work. Suggest the authors to use short sentences and professional English Editing.
Abstract should be re-written to make readers interested in this topic. It is better to define migraine as defined by IHCD-3 and make one reference rather than 7 refereces.
Comments on the Quality of English LanguageThis paper is useful for the reader. Needs extensive work. Our suggestion is to use short sentences and professional English Editing service to make the manuscript more readavble than now.
Abstract should be re-written to make readers interested in this topic. It is better to define migraine as defined by IHCD-3 and make one reference rather than 7 references.
Happy to review the revised version.
Reviewer 2 Report
Comments and Suggestions for Authors
Migraine is a complex disease, and the underlying mechanisms are not known. Although CGRP is one of the players implicated in pathogenesis, some subjects are refractory to blocking CGRP signaling, suggesting that other neuropeptides or other components may be involved. An overview of the function of neuropeptides in migraine pathogenesis and its treatment is provided in the present review.
The manuscript reports much information that may be useful to experts in the field, although a recent manuscript already reports and compares in detail the effects induced by PACAP and CGRP infusion in migraine subjects (doi: 10.1186/s10194-023-01569-2).
Below are reported some comments that the Authors should address before publication.
-The abstract is confusing; it should give a better idea of the topics covered.
-The purpose of the review should be better delineated.
-Please include a Methods section. It should include information such as the review period, the search engines that were used, and any other relevant details that will help readers.
-it would be helpful to include a table that summarizes the key findings of the studies evaluated, comparing the results of preclinical models and clinical studies.
-Please change the title of section 2.3. as the role of VIP in migraine pathogenesis is also discussed.
-In section 2.5. Clinical studies is reported that blockade of PACAP by a monoclonal antibody was effective in treating migraine attacks compared with blocking the receptor. The Authors should provide/suggest an explanation of these opposite results.
-The discussion section is repetitive and should be revised.
-Studies about blood levels of PACAP between patients and controls are contradictory, probably also because of methodological problems. Studies show that PACAP is involved in the pathophysiology of migraine, but its role as a therapeutic target to date is unclear. A clinical trial reports the efficacy of a monoclonal antibody against the peptide but not against its receptor. CGRP and PACAP infusion cause migraine-like headaches in about 2/3 of migraine patients. However, PACAP causes more premonitory symptoms and side effects than CGRP. These issues must be discussed in the text.
-The manuscript could benefit from a more critical assessment of the limitations of the studies reviewed, such as the discussion of the implications of these limitations for future research and clinical practice. This change would help readers to better understand the potential impact of the neuropeptide’s studies and their limitations on the field of migraine research and treatment.
Reviewer 3 Report
Comments and Suggestions for Authors
This paper should be an extensive review of recent discoveries about the role of PACAP in migraine pathogenesis that are leading to the discovering of new therapeutic options.
The topic is very important since a new era in migraine treatment started recently with anti CGRP targeting drugs (monoclonal antibodies and gepants) and probably will be enhanced with anti PACAP targeting molecules.
Unfortunately, there are many imprecisions in the abstract and in the introduction: tolerance over time of anti CGRP mAbs is not proven. Moreover, the fact that anti CGRP Ambs and gepants have low efficacy, low tolerability and many side effects is completely wrong. We need other migraine treatment but not for anti CGRP ineffectiveness but for the few patients who are non responders or have contraindications.
The work is not systematic: there are many repetitions in each paragraph and single topics are reported in a generic manner.
Table 3. There is a mistake (not anti CGRP trials but anti PACAP!)
Comments on the Quality of English Languagenone
Round 2
Reviewer 2 Report
Comments and Suggestions for Authors
The authors accepted most of my suggestions.
Reviewer 3 Report
Comments and Suggestions for Authors
The new version of the paper is slightly but not sufficiently improved:
Real life evidence had extensively demonstrated that anti CGRP targeting treatments are effective in much more than 50% of patients. I suggest to talk less of anti CGRP drugs in this review.
Moreover, when I said that the paper is not systematic I didn’t mean that the revision had to be systematic in term of method (I understood that it is a narrative review). I meant that the review is not presented in a sufficiently organized manner. For example, it should be re-organized for topics without repetitions of concepts in every subheading, introduction should be halved (many arguments are rewritten later).
Large place was given to VIP: I suggest to reserve less attention to VIP or extend the aim of the review and include VIP in the title
Comments on the Quality of English Languagenone
